# Do or Don’t: Results of a Multinational Survey on Antibiotic Prophylaxis in Urodynamics

**DOI:** 10.3390/antibiotics12071219

**Published:** 2023-07-22

**Authors:** Fabian P. Stangl, Laila Schneidewind, Florian M. Wagenlehner, Daniela Schultz-Lampel, Kaven Baeßler, Gert Naumann, Sandra Schönburg, Petra Anheuser, Susanne Winkelhog-Gran, Matthias Saar, Tanja Hüsch, Jennifer Kranz

**Affiliations:** 1Department of Urology, University Hospital of Bern, 3010 Bern, Switzerland; 2Department of Urology, University Medical Centre Rostock, 18057 Rostock, Germany; laila.schneidewind@med.uni-rostock.de; 3Clinic for Urology, Paediatric Urology and Andrology, Justus-Liebig-University Giessen, 35390 Giessen, Germany; florian.wagenlehner@chiru.med.uni-giessen.de; 4Southwest Continence Center, Schwarzwald-Baar-Klinikum, 78052 Villingen-Schwenningen, Germany; daniela.schultz-lampel@sbk-vs.de; 5Continence-and Pelvic-Floor-Centre, Franziskus-Hospital Berlin, 10787 Berlin, Germany; kaven.baessler@franziskus-berlin.de; 6Women’s Hospital, Helios Clinic Erfurt, 99089 Erfurt, Germany; gert.naumann@helios-gesundheit.de; 7University Women’s Clinic, Heinrich-Heine-University Düsseldorf, 40225 Düsseldorf, Germany; 8Department of Urology and Kidney Transplantation, Martin Luther University, 06108 Halle (Saale), Germany; sandra.schoenburg@uk-halle.de; 9Klinik für Urologie, Asklepios Klinik Wandsbek, 22043 Hamburg, Germany; p.anheuser@asklepios.com; 10Clinic for Urology and Pediatric Urology, St.-Antonius Hospital gGmbH, Academic Teaching Hospital of RWTH Aachen, 52249 Eschweiler, Germany; 11Department of Urology and Paediatric Urology, University Clinic RWTH Aachen, 52074 Aachen, Germany; msaar@ukaachen.de; 12Clinic for Urology and Pediatric Urology, University Medicine Johannes-Gutenberg-University Mainz, 55131 Mainz, Germany

**Keywords:** urodynamics, antibiotic prophylaxis, antimicrobial resistance, antimicrobial stewardship, guideline adherence

## Abstract

Antibiotic prophylaxis contributes substantially to the increase in antibiotic resistance rates worldwide. This investigation aims to assess the current standard of practice in using antibiotic prophylaxis for urodynamics (UDS) and identify barriers to guideline adherence. An online survey using a 22-item questionnaire designed according to the Checklist for Reporting Results of Internet E-Surveys (CHERRIES) was circulated among urologists and gynecologists in Austria, Germany, and Switzerland between September 2021 and March 2022. A total of 105 questionnaires were eligible for analysis. Out of 105 completed surveys, most responders (*n* = 99, 94%) regularly perform dipstick urine analysis prior to urodynamics, but do not perform a urine culture (*n* = 68, 65%). Ninety-eight (93%) participants refrain from using antibiotic prophylaxis, and sixty-eight (65%) use prophylaxis if complicating factors exist. If asymptomatic bacteriuria is present, approximately 54 (52%) participants omit UDS and reschedule the procedure until antimicrobial susceptibility testing is available. Seventy-eight (78%) participants do not have a standard procedure for antibiotic prophylaxis in their department. Part of the strategy against the development of bacterial resistance is the optimized use of antibiotics, including antibiotic prophylaxis in urodynamics. Establishing a standard procedure is necessary and purposeful to harmonize both aspects in the field of urological diagnostics.

## 1. Introduction

Urodynamics (UD) is an important diagnostic tool for assessing lower urinary tract dysfunctions [1]. Due to the requirement of pressure-measuring catheters in the bladder, it is an invasive procedure. Thus, urinary tract infections (UTIs) are the most common complication of urodynamic studies (UDS). However, the overall rate of UTIs after UDS is low, with reported rates of 2–3% [2]. Antibiotic prophylaxis has been shown to reduce the risk of bacteriuria, but not clinical UTIs after urodynamics. The European Guideline recommends omitting routine antibiotic prophylaxis in UDS based on the current evidence. However, daily clinical practice often deviates from the guideline recommendations by the lacking adherence to or availability or visibility of guidelines. Previous recommendations were primarily based on the paucity of data, but the current literature provides conclusive advice. Even UDSs on high-risk populations, e.g., patients with neurogenic lower urinary tract dysfunction and adjunct bladder emptying with indwelling or intermittent catheters, are at a considerably low risk of the development of a UTI after UDS [3]. The same finding applies to patients undergoing urodynamic testing with recurrent urinary tract infections. Albeit antibiotic prophylaxis does lower the rate of postinterventional UTI, the results failed to reach statistical and clinical significance, as shown in randomized studies and retrospective analysis [4,5]. Alternative approaches are equally evaluated and have shown promising results. Some phytotherapeutics may be beneficial in high-risk patients with the strict avoidance of antibiotic prophylaxis to treat postinterventional infections [6]. At this time, there are approx. 50.000 UDS treatments performed annually in Germany [7], reflecting the importance of adherence to current guidelines to decrease the antimicrobial resistance rates. Antimicrobial stewardship (AMS) is necessary to establish antibiotic-sparing strategies and regimens in all healthcare sectors [8]. The Infectious Disease Society of America published an updated definition of AMS in 2012, stating that ‘antimicrobial stewardship refers to coordinated interventions designed to improve and measure the appropriate use of antimicrobial agents by promoting the selection of the optimal antimicrobial drug regimen including dosing, duration of therapy and route of administration’ [9]. Implementing and promoting of AMS is a potent mechanism of infection prevention and is gaining relevance rapidly, especially in fields with high antibiotic consumption such as urology [10,11]. Constantly identifying and evaluating new targets and leverage points is of utmost importance [12]. Various studies exhibited the significant effect of AMS, especially when foregoing prophylaxis for minor procedures [13,14]. To promote AMS [8], the current investigation aimed to evaluate the current standard practice of care and to identify barriers to guideline adherence. Therefore, we conducted a multinational survey among gynecologists and urologists to identify the daily practice standards for antibiotic prophylaxis in UDS and gauge guideline adherence.

## 2. Results

A total of 105 questionnaires (Appendix A) were eligible for analysis; 60 (57%) were completed by urologists. The median number of urodynamics performed per week was five. Most participants (99 (94%)) routinely perform dipstick analysis prior to UDS, with 86 (86%) testing immediately before the UDS. Of those performing dipstick analysis, 27 (26%) used a microscopic urine examination. If urine sediment analysis is performed, most participants (75 (72%)) evaluate the urine samples on the day of urodynamics. Most colleagues do not routinely obtain a urine culture (68 (65%)). However, if dipstick analysis or urine sediment examination is suspicious for urinary tract infection, a urine culture is performed by 47 (45%) and 22 (21%) practitioners, respectively. The other indications for performing a urine culture were catheterized patients (13 (13%)) or those with neurogenic lower urinary tract dysfunction (10 (NLUTD; 10%)) (Figure 1). 

A total of 98 (93%) participants do not routinely use antibiotic prophylaxis for UDS. If antibiotics for prophylaxis in UDS are utilized, the participants use oral cephalosporins, sulphonamides, or nitrofurantoin in either a single-shot application or administration for up to five days. Sixty-eight (65%) participants prescribe antibiotic prophylaxis in distinct situations, such as catheterized patients (36 (35%)), patients performing intermittent self-catheterization (18 (18%)), those undergoing dipstick analysis (40 (39%)) or sediment examination (23 (22%)), those suspicious of a UTI without clinical symptoms, those with the presence of NLUTD (16 (16%)), and those with a post-void residual volume (PVR) (15 (15%)) (Figure 2). 

In the case of asymptomatic bacteriuria, 54 (52%) participants omitted UDS and awaited antimicrobial susceptibility testing. The occurrence of symptomatic post-procedural UTI was reported by 29 colleagues (28%). The absolute number of infectious complications ranges from 1 to 20 annually. 

In addition, specific antimicrobial stewardship programs (AMS) and their implementation were surveyed. Most responders have the recourse of using dedicated AMS teams (69%), whereas 20 (19%) are actively involved in those institutions. The majority of the responders (78 (78%)) have no standard operating procedures regarding antibiotic prophylaxis prior to urodynamics.

## 3. Discussion

We conducted a multinational questionnaire to gain insights into the current standard of practice in peri-procedural antibiotic prophylaxis in UDS. Most responders do not routinely prescribe antibiotic prophylaxis for UDS. However, approximately 65% of the responders prescribe routine antibiotic prophylaxis in particular situations, such as for indwelling catheter, NLUTD, post-void residual, and intermittent residual self-catheterization patients. Dipstick urine analysis on the day of the scheduled UDS without routine sediment analysis was most frequently performed. On the contrary, urine cultures are not performed regularly, but are reserved for patients with complicating factors, such as an indwelling catheter. 

In the case of asymptomatic bacteriuria (ASB), more than half of the responders omit UDS and prescribe antibiotic therapy according to antimicrobial susceptibility testing prior to UDS. The guidelines fail to offer guidance regarding infection screening before UDS. Taking urine cultures before UDS would offer a great level of safety for patients but increase the cost and administrative effort. Dipstick analysis often lacks sensitivity to clinical indifferent symptoms [15]. Infectious complications after UDS are reported in a vast range of patients, between 1.6 and 28.6%. The heterogeneity of data and their overall mediocre-to-poor quality impact clinical decision making negatively since robust current data from randomized trials are lacking [16,17,18,19]. Due to the conflicting evidence and overall low impact of peri-procedural prophylaxis, the EAU (European Association of Urology) Guideline on urological infections recommends omitting antibiotic prophylaxis for urodynamic studies [20]. Neither the EAU nor AUA (American Association of Urology) guidelines give recommendations regarding antibiotic prophylaxis in complex subgroups like NLUTD patients. The Society of Urodynamics, Female Pelvic Medicine, and Urogenital Reconstruction (SUFU) offer some guidance in their best practice policy statement, recommending prophylaxis in subgroups like NLUTD patients, but state that there is a small amount of evidence [21]. These statements have since been validated by Fox et al., who concluded NLUTD, elevated PVR, or the culmination of more than three risk factors, i.e., age or indwelling catheters, were the strongest predictors for post-interventional UTI [22]. The overall rate of UTI and morbidity in the named study was low. The number of randomized trials substantiating these recommendations is highly limited, and trials reporting a reduction of UTI post-UDS with antibiotic prophylaxis, e.g., Darouiche et al., are seriously underpowered, so these assumptions cannot be drawn [23]. There is no available evidence regarding the impact of asymptomatic bacteriuria and whether treating it before invasive testing is imperative. The available evidence is culminated in two studies by Benseler et al. and Foon et al., stating that there are no benefits to antibiotic prophylaxis [2,24]. Antibiotic prophylaxis treats post-procedural bacteriuria, but the pre-procedural data are limited. In the context of the technological evolution of urodynamic studies and the overall omitting of antibiotic prophylaxis in many procedures, which are significantly more invasive, a huge proportion of evidence needs to be updated. The authors of a few retrospective studies tried to evaluate risk factors for the development of UTI after UDS and drew similar conclusions. In general, NLUTD, indwelling catheters, and age are relevant risk factors for UTI [25].

AMS finds its way into general practice, where partakers do not usually collaborate. Overall, the participants have no standard operating procedure concerning antibiotic prophylaxis. Our survey uncovered considerable heterogeneity in the overall management of antibiotic prophylaxis in UDS. There is no general standard of care in the primary evaluation before planned diagnostic intervention and effective antibiotics before urodynamics. Additionally, the use of ASB before UDS deeply divides healthcare professionals, and the necessity of targeted prophylaxis is unfathomable. The same applies to the choice of antimicrobial agent in case of treatment, depicted by the broad spectrum of antibiotics used before UDS. 

Strict AMS guideline adherence is essential to unifying antibiotic prophylaxis regimens among urologists and gynecologists performing UDS. The collected data depict the level of diversity and divisiveness in the relevance of ASB and dosage of therapeutics, duration of prophylaxis, as well as the overall choice of agent. Unnecessary antibiotic prophylaxis therapies should be strictly avoided, and antimicrobial stewardship programs should be installed to provide low-threshold information and counselling concerning targeted therapy. 

Other studies have shown the benefit of AMS programs in lowering evolution pressure, especially for multi-drug-resistant organisms (MDRO) [26]. Inappropriate therapies could be reduced significantly by strict adherence to the guidelines, AMS programs, and interdisciplinary cooperation in in- and outpatient settings [27]. 

Interdisciplinary collaborations should be instigated at every level of healthcare. New processes to reduce antibiotics’ overall prescription and administration must be evaluated and implemented, and barriers that limit adequate therapy should be removed. The multidisciplinary approach is necessary to address multifaceted, complex interventions. AMS programs promote and monitor the correct usage of antibiotics, and subsequently, enable the reduction of the number of patients with MDRO. 

Failed antimicrobial stewardship is jointly responsible for the ongoing evolution of MDRO. The overuse of fluoroquinolones provides an excellent example of a failed AMS. Even though striking evidence exists concerning the use of fluoroquinolones, there is a lot of it [28]. On the other hand, implementing AMS programs delivers significant, timely results with lower resistance rates, and reduced prescribed antibiotic doses without forfeiting the patient’s safety or morbidity [29]. 

Clear recommendations are mandatory for greater guideline adherence, and standard operating procedures should be developed. Kranz et al. have shown crucial problems in guideline adherence, e.g., it contains ambiguous and complicated flowcharts and tables, and its conversion into clinical practice is difficult and impracticability [30,31]. To increase their feasibility, utilization, and implementation, the guidelines have to offer clear, quick, well-communicated, and accessible pathways for treatment options. 

The main goal and common ground should be encouraging strategies to tackle rising antibiotic resistance rates. The pre-emptive sparing of antibiotics in every utilization level is required to lower the burden on the evolution of antibiotic resistance. Ultimately, the main objective is avoidance of symptomatic UTI after UDS, which entails prolonged antibiotic therapy, increased costs and morbidity, and reduces the patient’s comfort and well-being.


**This study inherently has several limitations since we distributed a non-validated questionnaire survey. Furthermore, we gathered a relatively small, German-speaking, sample of urologists and gynecologists in Austria, Germany, and Switzerland. Therefore, we must consider the possibility of a relevant selection bias. There is a high likeability only physicians with interests in UDS and AMS attended the survey increasing the likelihood of a selection bias.**


## 4. Materials and Methods

### 4.1. Survey Development and Target Group

A multinational survey was conducted among specialists in urological infections according to the reporting guidelines for surveys found on the equator-network.org, an international initiative providing robust reporting guidelines [32]. The survey includes twenty-two items (fourteen multiple choice questions, five multiple choice questions with an additional text box to elaborate, and three open questions) generated via an explorative literature search on MEDLINE via PubMed using the MeSH terms “antibiotic prophylaxis, urodynamics, bacteriuria, urinary tract infection” to identify key questions concerning antibiotic prophylaxis in urodynamics. The survey was only available in German. All co-authors could suggest additional items if a unanimous decision among the authors was reached.

The target population was urologists and gynecologists performing urodynamics in Austria, Germany, and Switzerland. The goal was to have only one person per department participate in the survey.

### 4.2. Administration of the Survey

The survey was transferred to the online platform, SurveyMonkey^®^ (Survey Monkey^®^ by Momentive, CA, USA). It was tested for usability by all the co-authors before setting it up online. The questionnaire was available from the 1 September 2021 to the 1 of March 2022. An invitation was sent out via the Working Group “Urological Functional Diagnostics and Urology of Women” and “Infectiology and Hygiene” of the German Society for Urology e.V. and all the contacts of the co-authors. Altogether, six reminders were sent out.

The investigation was performed in accordance with the ethical standards of the institutional and national research committee and with the 1964 Helsinki Declaration and its later amendments or comparable ethical standards. Consent for this investigation was not mandatory, and no personal data were stored throughout the process. Survey participation was voluntary, and no incentives were offered. 

### 4.3. Statistical Analysis

Only complete questionnaires were analyzed. The Kolmogorov–Smirnov test was used to preliminarily assess the numeric distribution for each numeric variable. Descriptive statistics were made using the mean and standard deviation (SD) for normal distribution or with the median and interquartile range (IQR) for non-parametric data using SPSS Version 27.

## 5. Conclusions

There is a restricted level of agreement on the standard of care in UDS concerning the necessity of using antibiotic prophylaxis, the choice of agent, and the duration. Strong, elaborated, comprehensible, and well-disseminated guidelines are prerequisites for clarity and adherence. Strict AMS and standard operating procedures are essential in sparing antibiotics, tackling antibiotic resistance rates, and unifying the procedures for UDS. Given the common nature of genitourinary infections in general, urologists are at the forefront of challenging antibiotic overconsumption, especially when they use recommendations that are contrary to the guidelines. This survey offers new insights into the current standard of practice in peri-procedural antibiotic prophylaxis in UDS and depicts the often slow-paced penetration of guidelines into everyday practice.

## Figures and Tables

**Figure 1 antibiotics-12-01219-f001:**
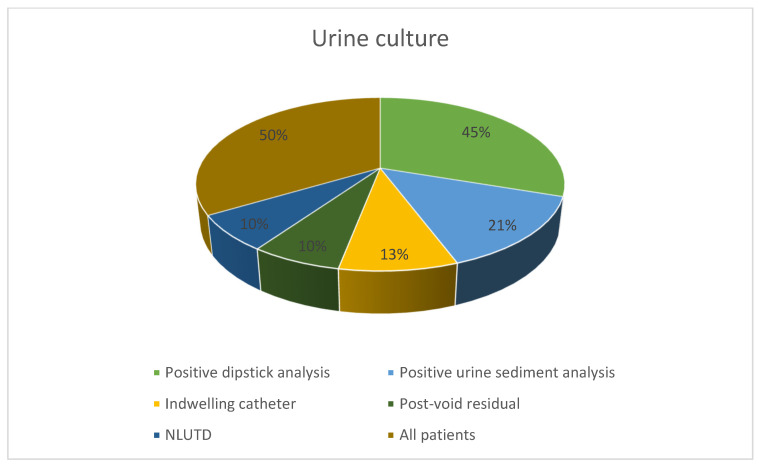
Reasons for performing a urine culture prior to urodynamics. Multiple answers were given.

**Figure 2 antibiotics-12-01219-f002:**
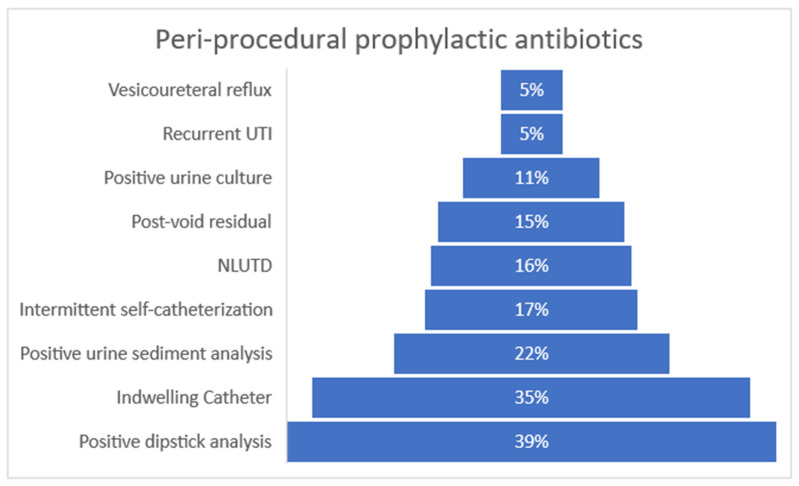
Reasons for antibiotic prophylaxis in urodynamics. Multiple answers were given.

## Data Availability

The acquired data in this study are available on request from the corresponding author. The data are not publicly available due to the General Data Protection Regulation by the European Union.

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
