# Peer review of "Do or Don’t: Results of a Multinational Survey on Antibiotic Prophylaxis in Urodynamics"

_antibiotics, 2023, doi:10.3390/antibiotics12071219_

Round 1
Reviewer 1 Report
The manuscript relates about important practical issues regarding antibiotic use in urodynamics.
The abstract, introduction, material and methods as the results, discussions and conclusions are clearly formulated, easy to follow with scientific formulation. The figures are concludent, maybe the colors of the first figure should be more different, to be easier to follow the results.
The references are ok, no self-citation was detected.
Author Response
Dear Reviewer 1
Thank you very much for your kind assessment of our manuscript. We edited the color scheme to increase readability, and thank you for your feedback.
Reviewer 2 Report
The article is correct under all formal requirements. I only have some comments. There is no such thing as a prophylactic antibiotics. Antibiotics can be divided into different categories, but there is no group of prophylactic antibiotics. There is prophylactic antibiotic therapy- as an action or antibiotic prophylaxis (if we want to use fewer words). So please use the correct terminology. I also suggest changing the beginning of the title. "Do or Don´t" phrase suggests that You will answer the question whether or not to use prophylactic antibiotic therapy in UDS. But the article is about the question "Is antibiotic prophylaxis applied or not?". One more little note, please enter an explanation of the abbreviation UDS- Urodynamic studies.
Author Response
Dear Reviewer 2
Thank you very much for your kind assessment. Your feedback really helped to improve our manuscript. We changed the terminology to the correct term “antibiotic prophylaxis” and agree with you- scientific language needs to be concise and you uncovered a major flaw of our manuscript.
We included the abbreviation for UDS in the first paragraph. Concerning the title of our manuscript you are correct once more- we do not answer the question about “do or don´t”, but we felt a “catchphrase” in the title would attract readers, and therefore serve our purpose of endorsing antimicrobial stewardship.
Once again, we would like to express our gratitude for your important and applicable remarks.
Reviewer 3 Report
Compliance with regards to antimicrobial stewardship, particularly for urinary tracts infections is highly variable, but not well documented. This group performed an international survey to determine practices for prevention of urinary tract infection associated with urodynamic analyses. The results are interesting and provide evidence for the general practices of urologists and urogynecologists in Germany, Austria and Switzerland. The results are clearly presented and provide direct insight into clinical practices.
Minor concerns:
1. Can the authors please indicate the language that the survey was available to the respondents and could this be another potential limitation.
2. Although the focus of the manuscript is on the antibiotic usage, data was collected on the screening process which is also a concern in the field. Can the authors please comment on the appropriateness of the use of the dipstick and other screening methods according to the current guidelines?
Author Response
Dear Reviewer 3
Thank you very much for your kind assessment of our manuscript.
- You raised an important issue concerning the language of our survey. We included the information and edited the limitation section.
- Currently there are no specific recommendations regarding screening for infection. The safest way to preclude infection would be culturing urine before invasive testing. But decreased practicality and increased cost often foreclose the safest option. Mostly, dipstick analysis is used to screen for infection. We added a section in the discussion to address the issue further.
We would like to thank you again for your valuable input and feel your feedback really helped improving our manuscript.